# Multi-Scale Feature and Multi-Channel Selection toward Parkinson’s Disease Diagnosis with EEG

**DOI:** 10.3390/s24144634

**Published:** 2024-07-17

**Authors:** Haoyu Wu, Jun Qi, Erick Purwanto, Xiaohui Zhu, Po Yang, Jianjun Chen

**Affiliations:** 1Department of Computing, Xi’an Jiaotong-Liverpool Univeristy, Suzhou 215000, China; haoyu.wu18@student.xjtlu.edu.cn (H.W.); erick.purwanto@xjtlu.edu.cn (E.P.); xiaohui.zhu@xjtlu.edu.cn (X.Z.); 2Department of Computer Science, The University of Sheffield, Sheffield S10 2TN, UK; po.yang@sheffield.ac.uk

**Keywords:** machine learning, Parkinson’s disease, EEG, channel selection, wavelet packet transform

## Abstract

Objective: Motivated by Health Care 4.0, this study aims to reducing the dimensionality of traditional EEG features based on manual extracted features, including statistical features in the time and frequency domains. Methods: A total of 22 multi-scale features were extracted from the UNM and Iowa datasets using a 4th order Butterworth filter and wavelet packet transform. Based on single-channel validation, 29 channels with the highest R2 scores were selected from a pool of 59 common channels. The proposed channel selection scheme was validated on the UNM dataset and tested on the Iowa dataset to compare its generalizability against models trained without channel selection. Results: The experimental results demonstrate that the proposed model achieves an optimal classification accuracy of 100%. Additionally, the generalization capability of the channel selection method is validated through out-of-sample testing based on the Iowa dataset Conclusions: Using single-channel validation, we proposed a channel selection scheme based on traditional statistical features, resulting in a selection of 29 channels. This scheme significantly reduced the dimensionality of EEG feature vectors related to Parkinson’s disease by 50%. Remarkably, this approach demonstrated considerable classification performance on both the UNM and Iowa datasets. For the closed-eye state, the highest classification accuracy achieved was 100%, while for the open-eye state, the highest accuracy reached 93.75%.

## 1. Introduction

Parkinson’s disease (PD) is a chronic and progressive neurodegenerative disorder that affects millions of people worldwide [1]. It is characterized by various motor and non-motor symptoms, thus making it a complex condition to diagnose and manage. The cardinal motor symptoms of Parkinson’s disease include bradykinesia (slowness of movement), rigidity (stiffness of muscles), tremors, and postural instability. However, PD is not solely a motor disorder. Patients may also experience a wide range of non-motor symptoms, such as depression, anxiety, sleep disturbances, cognitive impairment, and autonomic dysfunction, as well as some non-motor symptoms, such as sleep disturbances, may appear in the early stage of PD [1,2]. To date, the diagnosis of PD is mainly based on clinical assessment and the presence of specific motor symptoms. There is no definitive diagnostic test for PD, so healthcare professionals must carefully evaluate the patient’s history, perform a thorough physical examination, and rule out other conditions that may mimic PD symptoms. Motivated by Health Care 4.0, which is a new era of health care propelled by the advent of Industry 4.0 [3], numerous researchers have explored the potential of utilizing wearable sensors signal [4] for the computer-based diagnosis of Parkinson’s disease, such as using an accelerometer [5,6] and EEG. EEG testing, a non-invasive and cost-effective technique widely available in medical centers that captures the electrical and magnetic field signals generated by neuronal activity on the scalp at specific frequencies [7], has demonstrated effectiveness in diagnosing and predicting various neurological disorders, including epileptic seizures, the development of biomarkers for Alzheimer’s disease, and the detection of abnormalities in schizophrenia [8,9,10].

Nevertheless, EEG signals present two main challenges for analysis: their stochastic nature and low signal-to-noise ratio (SNR) [11]. The low SNR indicates a high noise level within EEG signals, thus complicating pre-processing efforts, as current algorithms partially filter out the signal while attempting to remove noise. This adversely impacts classification model performance. Moreover, the inherent randomness of EEG signals demands the use of sophisticated nonlinear and dynamic models for effective feature extraction and classification, leading to considerable computational time and resource requirements. To address the low SNR issue, several advanced pre-processing algorithms have been developed. For example, the research by [12] provided evidence of the effectiveness of their Artifact Subspace Reconstruction (ASR) as an automate method for removing artifacts. This study suggested that a cutoff parameter between 20 and 30 is preferable, as opposed to the previously suggested and default values of 5 to 7, which resulted in excessive removal of brain activities. And it also revealed that ASR enhances the quality of ICA decomposition, as indicated by an increased number of dipolar independent components. Gu et al. [13] have also proposed an automatic ocular artifact removal (AOAR) method for EEG signals, which outperformed the other methods in terms of the root mean square error, SNR, and correlation coefficient, particularly in cases with lower SNR levels. Islam et al. [14] made a good review of those studies in this field.

Considering that the classification models remain complex due to the high dimensionality of the feature vector, three approaches are commonly employed to reduce dimensionality: feature selection, frequency band analysis, and channel selection. Feature selection primarily relies on methods such as single-factor analysis of variance or chi-square analysis, thereby often employing *p*-value calculations [15,16]. Alternatively, deep learning techniques can be employed to fuse the feature matrix [17]. Frequency band analysis focuses on examining different frequency rhythms to identify more informative patterns. Similar to the aforementioned methods, channel selection aims to identify the most informative EEG channels to reduce the dimensionality of the feature vector at the channel level.

To reduce the dimensionality of traditional EEG features based on manual extraction (such as statistical features in the time and frequency domains), we propose a channel selection approach using single-channel classification, thus building upon previous research [18]. We selected a common set of 29 channels for our analysis based on the montage utilized in two publicly available datasets used in the referenced study. The specific contributions are outlined as follows:A Parkinson’s disease EEG classification model depending on the public UNM and Iowa datasets has been proposed, which uses multi-scale features: a time domain, frequency domain features extracted by band-pass filter and wavelet packet transform, and an entropy feature (Section 3);We introduced a channel selection method consisting of 29 channels, which achieved a remarkable recognition accuracy of 100% on the UNM training dataset. Furthermore, the model successfully passed the out-of-sample testing on the Iowa dataset, thus demonstrating its generalizability. (Section 4 and Section 5);

This paper follows the below structure: In Section 2, several state-of-the-art studies relevant to the domain of this paper will be discussed. In Section 3, we will discuss the detailed implementation processes and methods for the dataset, data pre-processing, and feature engineering. In Section 4, we will delve into the channel selection based on single-channel classification. In Section 5 and Section 6, we will present the classification results, including the validation of the proposed channel selection method on the training dataset and the out-of-sample testing, as well as a comparison of the classification performance before and after channel selection.

## 2. Related Works

Gulay et al. [19] introduced a hybrid feature extraction method called EEMD VAR for EEG signals, thus integrating the EEMD and VAR techniques. Empirical comparisons were made between the EEMD VAR method, Hjorth parameters, and the AR model. The proposed algorithm achieved a maximum classification accuracy of 100%, thereby significantly outperforming the maximum accuracy of 72% attained by the AR method and Hjorth parameters. This highlights the potential of the EEMD VAR method to provide robust classification performance. Furthermore, the results show superior overall diagnostic accuracy, sensitivity, specificity, and AUC values for subjects with Parkinson’s disease (PD). Among the classification algorithms evaluated, the ANN stands out as the most prominent based on all performance evaluation metrics considered in this study. One limitation of this study lies in the lack of discussion or exploration of channel selection for EEG signals. The researchers did not explain the reason for the channel selection and running of the out-of-sampling test, which raises concerns about the reliability of their channel selection approach. The use of the ANN significantly increases the computational costs associated with feature extraction and model training as well.

The study of Coelho et al. [20] examined the effectiveness of Hjorth features extracted from electroencephalographic (EEG) signals as potential biomarkers for Parkinson’s disease (PD). Biomarkers are measurable indicators of some biological state or condition. For Parkinson’s disease, typical examples include α-synuclein, uric acid levels, mutations in relevant genes such as LRRK2 or GBA, and brain MRI, which have been widely applied in Parkinson’s disease risk assessment and symptom diagnosis [21]. Currently, the research on AI-based Parkinson’s disease diagnosis using EEG is still exploring the use of machine learning features as potential biomarkers. The focus has primarily narrowed to EEG frequency bands associated with Parkinson’s disease symptoms, such as alpha, theta, and beta bands [22,23,24]. Notably, the prominent role of the beta band in diagnosing resting tremors suggests that features related to the beta band could serve as new PD biomarkers. Their analysis utilized EEG data from PD patients exposed to auditory stimuli, which were obtained from the publicly available database known as The Patient Repository for EEG Data + Computational Tools (PRED + CT). The investigation revealed notable differences in the proposed biomarkers across various brain lobes, including parietal, frontal, central, and occipital regions, as well as between healthy individuals and those with PD. Support Vector Machine (SVM), Random Forest, and K-Nearest Neighbors (KNN) algorithms were employed for the classification task, along with a five-fold cross-validation process. The proposed method achieved a high accuracy of 89.56% in distinguishing PD patients from healthy individuals using an SVM classifier. However, the study lacks research on channel selection, because although it identified some channels that exhibited outstanding performance on different features, it did not further conduct training and validation for channel selection.

Smrdel [25] employed both common spatial patterns (CSPs) and a Laplacian mask to facilitate robust selection and feature extraction in their study. They utilized the CSP whitening matrix to identify the channels that exhibited the greatest potential for distinguishing between EEG signals from healthy individuals and those with PD. By leveraging the selected features obtained through CSPs, they achieved a classification accuracy of 85% when categorizing EEG records into groups of healthy controls and PD patients. Furthermore, using features extracted via the Laplacian operator, they achieved a classification accuracy of 90%; however, their research lacks validation of the channel selection approach. Although they identified some channels with the highest information content, they failed to demonstrate whether this channel selection approach can prevent overfitting when applied to datasets beyond their own.

Dar et al. [26] introduced a novel architecture, 1D-CRNN-ELM, which merges an Extreme Learning Machine (ELM) with a one-dimensional Convolutional Recurrent Neural Network (1D-CRNN) for emotion detection in patients with Parkinson’s disease (PD). This hybrid approach leverages the strengths of both neural networks to enhance the feature extraction and classification of EEG data. This framework is designed to handle various emotions and experimental conditions, thus enabling cross-dataset learning. After pre-processing the EEG data, the CRNN serves as a feature extractor, while the ELM functions as the classifier. The CRNN can be fine-tuned with additional datasets to learn new emotion sets. The study utilizes cross-dataset learning by training on PD patient data and fine-tuning with publicly available datasets and vice versa. Using an 80:20 train–test data split, the model achieved high accuracy values. Using the SEED-IV dataset to fine-tune the architecture resulted in an accuracy of 92.5%. Leave-one-out cross-validation yielded high mean accuracy values for all datasets in the experiment. The study also demonstrated that high-performance emotion detection is feasible using only 1-second EEG segments from 14 channels. However, the CRNN’s computational costs remain high, and despite extensive validation within their dataset, cross-dataset validation is still lacking. Additionally, the reliability of the ELM is questionable.

Numerous studies have been conducted on the publicly available New Mexico and Iowa datasets, some of which serve as the foundation for this research. A new LEAPD index was created by Anjum et al. [27], which effectively and quickly distinguishes individuals with Parkinson’s disease (PD) from control subjects. The LEAPD method encodes the power spectral density (PSD) using a limited number of parameters and holds promise for generating PD diagnostics or enhancing control algorithms for real-time applications. Moreover, it has the potential to improve the accuracy of predicting motor progression and refining classifications of PD-related sub-types. The utilization of the LEAPD index can contribute to the development of cost-effective diagnostic tools and real-time control signals for Parkinson’s disease and other neurodegenerative conditions. Levodopa is a medication commonly used for the treatment of Parkinson’s disease [4,28]. However, it remains unclear whether the channel selection method based on the LEAPD feature would be effective when applied to traditional multi-scale features. Furthermore, the study did not explore the channel selection scheme based on these traditional features within the dataset. In our study, we employed single-channel evaluation for channel selection. Another study focusing on the detection of abnormal EEG signals has provided insights into the extraction of sub-band wavelet coefficients and the calculation of statistical characteristics using the Discrete Wavelet Transform (DWT) [29]. This study extracted six statistical features from the wavelet coefficients of each sub-band. Additionally, this study has addressed the issue of feature redundancy in multi-scale feature extraction through the utilization of ensemble learning techniques.

## 3. Machine Learning-Based Classification Model for Parkinson’s Disease Patients’ EEG Signal

The classification model was developed based on the UNM dataset and tested on the UNM dataset and Iowa dataset. The implementation structure of this model is illustrated in Figure 1. The UNM dataset was divided into a training set and a validation set, while the Iowa dataset was utilized as an external validation data source. After undergoing pre-processing, all data were subjected to two feature extraction approaches, thus resulting in the generation of 13 and 9 distinct features, respectively. These 22 features constituted the feature vector used to train six commonly used machine learning models. The model’s classification performance and the reliability of the channel selection approach were evaluated based on the obtained results.

### 3.1. Datasets

We utilized electroencephalogram (EEG) recordings obtained from two separate studies conducted at the University of New Mexico (UNM) in Albuquerque, New Mexico, and the University of Iowa (UI) in Iowa City, Iowa [22,27,30]. The UNM dataset consisted of EEG recordings from 27 patients diagnosed with Parkinson’s disease (PD) and 27 control participants (a control record was excluded from the current study due to its insufficient duration), while the UI dataset included EEG recordings from 14 PD patients and 14 control participants. For the PD patients from the UNM dataset, EEG recordings were obtained during the OFF medication sessions, specifically in the defined OFF levodopa period, which occurred 12 h after the last dose of dopaminergic medication.

As discussed by Anjum et al. [27], the control participants were carefully matched with the PD patients in terms of age and sex, and there were no significant differences in education or pre-morbid intelligence measurements between the two groups. Parkinson’s disease patients underwent neuropsychological and questionnaire assessments in their ON state, with a neurologist administering the United Parkinson’s Disease Rating Scale (UPDRS) motor scores. Resting state EEG recordings were acquired from UNM subjects in both eyes-open and eyes-closed conditions, while EEG data from the 28 Iowa subjects were recorded solely in the eyes-open condition. The details are shown in Table 1.

### 3.2. Pre-Processing

The EEG signals underwent a filtering process ranging from 0.1 Hz to 100 Hz, with a sampling rate of 500 Hz, using two different 64-channel montages. In the UNM dataset, the online reference was set to channel CPz, while in the Iowa dataset, it was set to channel Pz. Consequently, after removing channels that exhibited differences across those two montages, 59 channels were utilized for subsequent feature extraction and classification. To address artifacts caused by eye blinks, independent component analysis (ICA) was employed, thus resulting in the extraction of 56 independent components [31,32]. Additionally, a 60 Hz notch filter was applied to eliminate interference from the power line.

For the UNM dataset, each EEG recording comprised two distinct sessions: one with eyes closed and another with eyes open. Following the feature extraction process, the features corresponding to these two events were identified and segregated into two separate datasets.

The two events, namely eyes closed and eyes open, had a duration of 60 s each, thus resulting in 30,000 samples for each event within a single recording. To facilitate further analysis, both events were divided into 15 segments, with each segment containing 2000 samples. Within each segment, we extracted six sub-bands to enable additional feature engineering. These sub-bands included delta (1–4 Hz), theta (4–8 Hz), alpha1 (8–10 Hz), alpha2 (10–13 Hz), alpha (8–13 Hz), and beta (13–30 Hz). The boundary frequencies were treated as separate sub-bands. To separate these sub-bands, we employed two methods: a 4th order Butterworth IIR filter and Wavelet Package Transform. From each sub-band, we extracted multi-scale features, thus capturing diverse aspects of the EEG signal.

### 3.3. Feature Extraction

In order to extract multi-scale features from the aforementioned six sub-bands, both IIR filtering and Wavelet Packet Transform were applied to extract the corresponding frequency ranges from the original signal, as shown in Figure 2. We employed a 4th Butterworth IIR filter as a band-pass filter and a notch filter to generate sub-bands and extract 13 features. The purpose of the notch filter was to eliminate noise from the power frequency signals at 60 Hz and 180 Hz. Next, we extracted the six sub-bands mentioned earlier from each pre-processed segment by utilizing the IIR filter as six corresponding bandpass filters. From each sub-band, we extracted a total of 13 features, thus consisting of 9 time domain features, three frequency-domain features, and one entropy feature, as shown in Table 2, where MAD represents for the mean value of the signal’s absolute deviation, which is expressed as below:(1)$$MAD=\frac{1}{m}\sum_{i}^{m}\left|x_{i}-u\right|\;\;i=1,2,\cdots ,m$$
where $x_{i}$ represents the *i*th data point in the sequence, while *u* represents the arithmetic mean value of the sequence. RMAV represents for the ratio of the absolute mean value between two sub-bands. To derive the remaining nine features, we employed Wavelet Packet Transform using the db5 wavelet. For each segment, we utilized the pre-processed signal without any resampling. In order to generate the 6 sub-bands, we set the decomposition level to six, thus taking into consideration the Sampling Theorem. Specifically, we selected six coefficients: AAAAAA6, AAAAAD6, AAAADA6, AAAADD6, AAAAD5, and AAAD4, as illustrated in Figure. From these coefficients, we calculated nine features, as outlined in Table 3.

As was mentioned before, there are two kinds of eye state in UNM dataset: eyes open and eyes closed. Considering the significant differences in data collected based on these two eye conditions and the non-parallel nature of the data collection process, we applied our feature engineering separately to these two types of data, thus resulting in corresponding feature sets, as shown in Table 4. For each participant with one eye state and one feature extraction method, the dimensionality of the feature vector came out to 59 × 15 × 6 × 9 or 13. Here, 59 represents the common number of channels, 15 denotes the number of segments, 6 indicates the number of sub-bands extracted, and 9 or 13 signifies the number of extracted features depending on different feature extraction methods.

## 4. Channel Selection

Channel selection is one of the main contributions of this study. Our channel selection was based on the UNM dataset, where the training set was used to train a simple SVM model, and the validation set was used to evaluate the classification performance of each channel. Figure 3 and Figure 4 demonstrate the classification performance of each channel after feature extraction using Wavelet Packet Decomposition. It is worth noting that some channels shown in the images do not exist in the Iowa dataset and were ignored in subsequent testing. The evaluation metric for assessing the classification performance of individual channels is the R2 score, and we set the threshold for channel selection at 0.7, thus meaning that channels with an R2 score below 0.7 were not accepted. Following this criterion, a total of 29 channels that met the requirement were selected based on the two feature extraction methods. For the feature set derived from the 4th Butterworth IIR filter, 11 channels were selected: ‘Oz’, ‘F4’, ‘P8’, ‘CP2’, ‘Cz’, ‘Fp2’, ‘P2’, ‘FC5’, ‘T7’, ‘O1’, and ‘FC6’. For the feature set derived from Wavelet Packet Decomposition, 22 channels were selected: ‘TP7’, ‘TP9’, ‘TP10’, ‘FC3’, ‘FC4’, ‘FC5’, ‘PO7’, ‘PO8’, ‘CP3’, ‘CP5’, ‘P1’, ‘P3’, ‘P4’, ‘P5’, ‘P6’, ‘P8’, ‘AF3’, ‘C4’, ‘F5’, ‘Oz’, ‘O1’, and ‘O2’. That is, 29 channels were selected in total. Figure 5 shows an example of the relationship between 20 selected channels’ feature values and model outputs. In order to visualize this relationship into a 2D figure, we simply took the average value of all feature values for each channel, as the shown feature value. SHAP (SHapley Additive exPlanations), which is a method rooted in game theory that elucidates the results of machine learning models, was applied by using a SHAP Python tool to describe the impact on model output for every feature value in this figure [33]. Therefore, positive correlation in this diagram can be observed as high feature values accompanied by positive SHAP values, and all these 20 channels showed a positive correlation with the model output.

## 5. Results

In the multi-channel classification task, six machine learning models were employed: Logistic Regression (LR), KNN, SVM, AdaBoost, XGBoost, and Random Forest. Logistic Regression had a maximum iterate number of 5000, while KNN used seven neighbors. The linear kernel was applied to SVM, just like in the single-channel evaluation task. The learning rates for AdaBoost and XGBoost 1.5.1 were both set to 0.01. AdaBoost terminated boosting at a maximum of 50 estimators, while XGBoost had a gamma value of 0.05. The Random Forest model had a default setting of 100 trees. Considering the objectives of this study, as well as the complexity and limited interpretability inherent in deep learning, we have not devised a comprehensive deep learning EEG classification framework. Instead, a simple neural network with two fully connected (FC) layers with layer sizes of 512 and 1 was employed as a classifier. The dropout rate was set to 0.2 after the first FC layer, and we chose binary cross entropy as the loss function; the optimizer was Adam, as usual. All the parameters were selected by applying a grid search. For training and testing, data with features from all channels and the best channels were used separately. The results were presented and compared in our previous study [18]. In this session, we will briefly give the overall evaluation results on the UNM dataset, which are shown in Table 5 and mainly focus on the channel selection results and the out-of-sample testing.

We conducted testing on the evaluation set of the UNM dataset and calculated the R2 score, which is shown in Figure 6. The possibility of selecting fewer channels to significantly reduce the dimensionality of the features was also evaluated by comparing the classification performance of two feature extraction methods, along with their respective channel selection schemes and the intersection of these schemes. The experimental results, presented in Figure 7, provide insights into the assessment of selecting fewer channels to achieve a greater reduction in feature dimensionality.

Due to the availability of only open-eye state data in the Iowa dataset, we designed two training schemes for out-of-sample testing: one using only open-eye state data from the UNM dataset as the training set and the other using both eye states from the UNM dataset. Both schemes utilized the Iowa dataset as the test set, and the results are presented in Table 6. When using the UNM (open-eye) dataset as the training set and the Iowa (open-eye) dataset as the test set, the LR model performed the best, thus correctly classifying 24 out of 28 samples, followed by the SVM and other models, with 23 out of 28 samples correctly classified. All models showed better performance compared to the results obtained from testing within the UNM dataset, with an average increase in accuracy of approximately 0.03 and a negligible average decrease in the R2 score of less than 0.01.

Furthermore, the results obtained by combining open-eye and closed-eye samples from the UNM dataset for training showed improvements compared to using only open-eye data. The average increase in accuracy was approximately 0.01, while the average decrease in sensitivity was around 0.05. On the other hand, the average increase in specificity was approximately 0.07, and the average increase in the area under the curve (AUC) was around 0.01. Additionally, the average increase in the R2 score was approximately 0.02.

## 6. Discussion

Our findings, as shown in Figure 6, indicate that the dataset with channel selection achieved a higher average R2 score compared to the dataset with all channels for the seven classifiers examined. Specifically, the KNN, XGBoost-linear, and Random Forest models showed improvements, while the AdaBoost and XGBoost-Tree models exhibited noticeable decreases in performance. However, when considering the overall picture, the classification models without the application of channel selection demonstrated a relatively poorer fit on the test set but showed better fit on the training set. This suggests that the utilization of all channels during training led to the occurrence of overfitting, thus resulting in inferior performance on the test set. These findings also demonstrate the successful implementation of our channel selection approach, which effectively balances the need for excellent classification performance (accuracy, sensitivity, precision, etc.) while mitigating the overfitting associated with involving all channels in the classification process.

Considering further selection of the chosen channels, we took the intersection (4 channels) and union (29 channels) of the channel selection results based on traditional multi-scale features (11 channels) and statistical features of the wavelet coefficients (22 channels). We then performed cross-validation on the training set to obtain the average R2 score and compared it with the R2 score on the test set. According to Figure 7 and Figure 8, it was observed that using the 11 channels effectively reduced the R2 score on the training set while maintaining a better accuracy and R2 score on the test set, whereas the 22 channels did not exhibit the same behavior. Subsequently, testing the intersection (four channels) revealed a significant decrease in the R2 score on the training set, but the accuracy, sensitivity, and R2 score on the test set exhibited excellent performance, thus averaging above 0.9. This demonstrates that among the chosen 29 channels, the four channels ‘Oz’, ‘P8’, ‘FC5’, and ‘O1’ primarily contributed to the observed effects (although the 11-channel scheme exhibited better classification accuracy on the test set). The accuracy and R2 score reflect the classification performance and fitting degree on the respective test sets, and higher values indicate better model performance. According to Figure 8, it is evident that the 29-channel selection scheme remains the most reliable, with the majority of results concentrated above an R2 score of 0.7 and an accuracy of 0.8. Moreover, the R2 score exhibited minimal fluctuations in the high accuracy (0.9) region, thus consistently hovering around 0.8 and 0.9 and indicating good generalization ability of this scheme. Further testing can be conducted on out-of-sample data. In contrast, the four-channel selection scheme clearly exhibited severe overfitting, and therefore, this scheme was not considered for subsequent out-of-sample validation. It is worth noting that the 11-channel scheme showed similar performance, with an R2 score above 0.6 and an accuracy above 0.7. Although it is not as outstanding as the 29-channel scheme, it can be considered as an alternative depending on the specific application. However, it showed a larger fluctuation range in the high accuracy (0.9) region, thus indicating poorer generalization ability.

From our out-of-sample testing results, it is evident that the proposed channel selection scheme has passed the external testing phase and exhibited comparable or even better performance than the original internal testing conducted on the dataset. Moreover, we observed that incorporating both open-eye and closed-eye samples in the training set was beneficial for classifying EEG signals under the open-eye state. We speculate that this improvement can be attributed to the fact that EEG signals recorded during the closed-eye state are purer, with less noise compared to signals obtained during the open-eye state. Participants also experience less interference during the closed-eye state, thus making the signals closer to the theoretically pathological original signals and more valuable as references. In other words, training the machine learning model with a combination of closed-eye and open-eye datasets helps the model consider the features of both scenarios comprehensively and adjust the confidence intervals to approach the real range of Parkinson’s disease onset. This, in turn, improves the classification of signals under the eyes-open condition.

## 7. Conclusions

EEG has proven to assist in the clinical diagnosis of Parkinson’s disease. However, the computational costs associated with feature engineering and machine learning training on EEG signals are substantial, which hinders the monitoring and assisted diagnosis of Parkinson’s disease in uncontrolled environments such as homes. Numerous studies have been conducted in this direction. In this paper, we conducted a study utilizing the publicly available UNM dataset and proposed a Parkinson’s disease EEG classification model based on machine learning and traditional feature extraction techniques. We effectively reduced the feature dimensionality by employing a single-channel selection approach, thus theoretically reducing it by more than half. The proposed channel selection scheme not only achieved a maximum classification accuracy of 100% within the UNM dataset but also demonstrated its generalization capability through out-of-sample testing using the Iowa dataset. This verifies the effectiveness and generalizability of our channel selection method. In the future, we aim to develop an automated EEG channel selection model utilizing deep learning techniques based on our current methods. This model is designed to provide tailored channel selection schemes for individual Parkinson’s patients. Additionally, we anticipate extending this approach to other EEG-related tasks, such as epilepsy diagnosis and emotion recognition, thereby enhancing its applicability and utility in various clinical and research settings. We believe that this channel selection approach can provide valuable insights for future research in related fields.

## Figures and Tables

**Figure 1 sensors-24-04634-f001:**
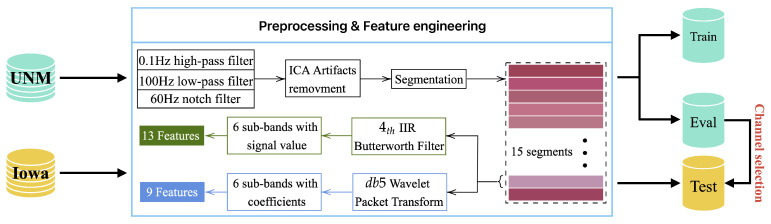
The implementation structure of the proposed EEG classification model.

**Figure 2 sensors-24-04634-f002:**
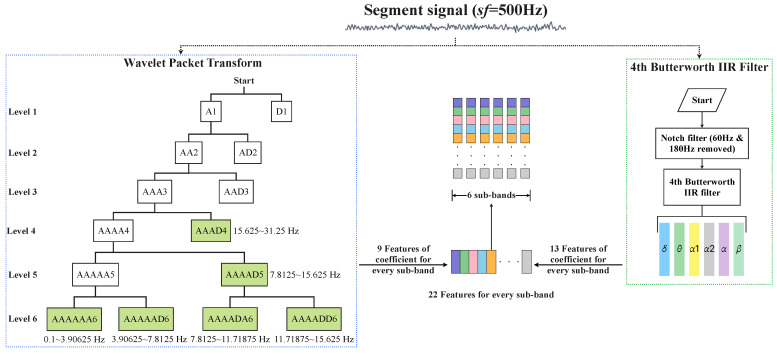
Feature engineering structure for one segment signal, where features obtained from different methods are stored separately.

**Figure 3 sensors-24-04634-f003:**
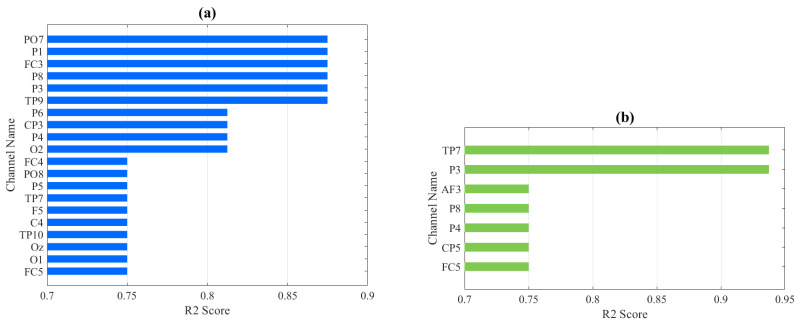
Single-channel evaluation R2 score result for eyes-closed (**a**) and eyes-open (**b**) feature dataset extracted by Wavelet Packet Transform.

**Figure 4 sensors-24-04634-f004:**
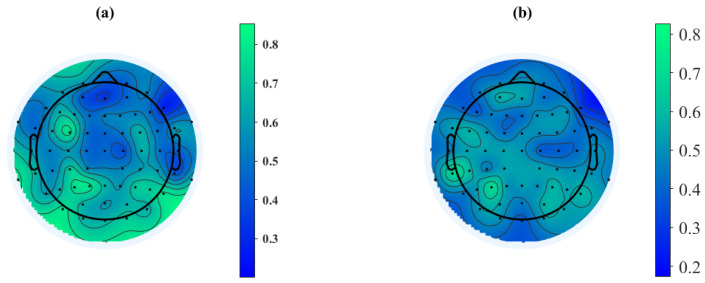
Single-channel evaluation brain maps for eyes-closed (**a**) and eyes-open (**b**) feature dataset extracted by Wavelet Packet Transform. The color bar refers to the R2 score, and high R2 score means high single-channel classification performance.

**Figure 5 sensors-24-04634-f005:**
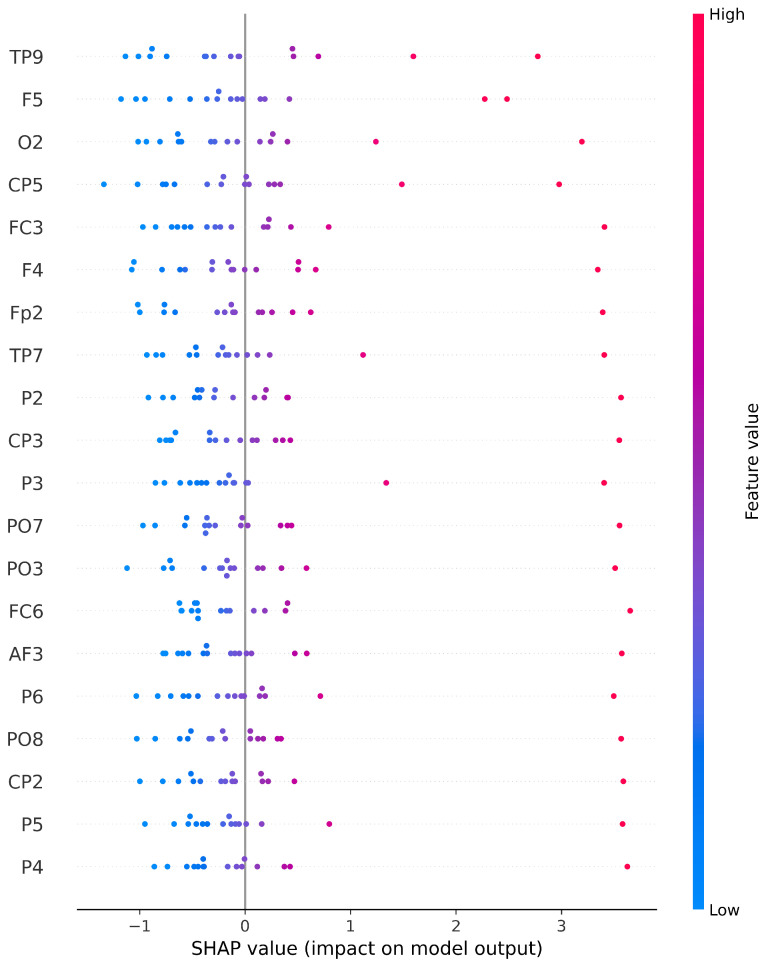
Relationship between channel feature value and the model output.

**Figure 6 sensors-24-04634-f006:**
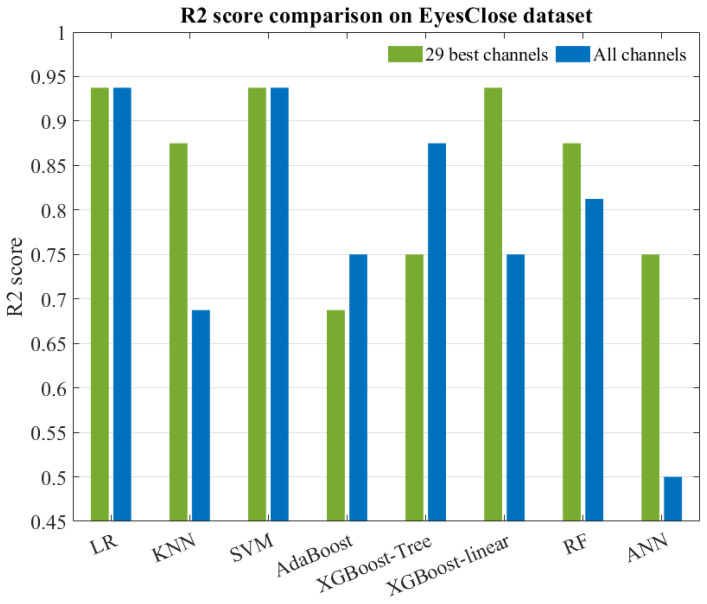
R2 score comparison on eyes-closed evaluation set.

**Figure 7 sensors-24-04634-f007:**
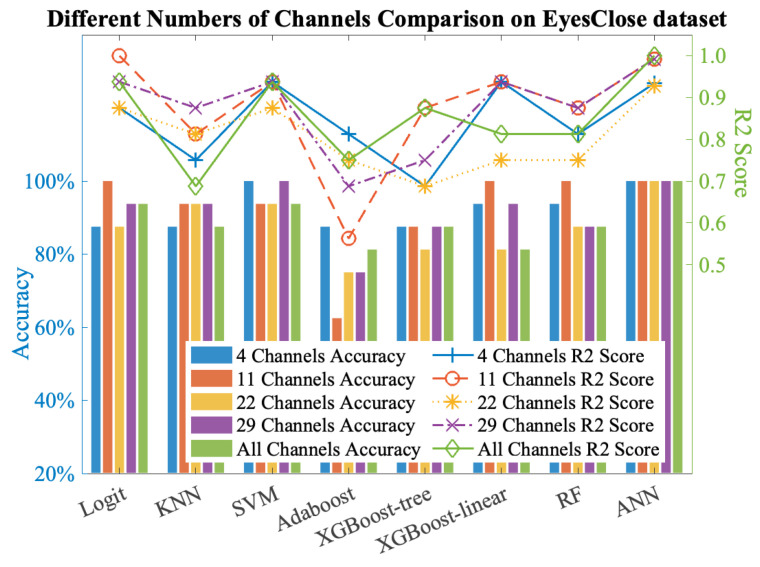
Comparison between different channel selection schemes on eyes-closed dataset.

**Figure 8 sensors-24-04634-f008:**
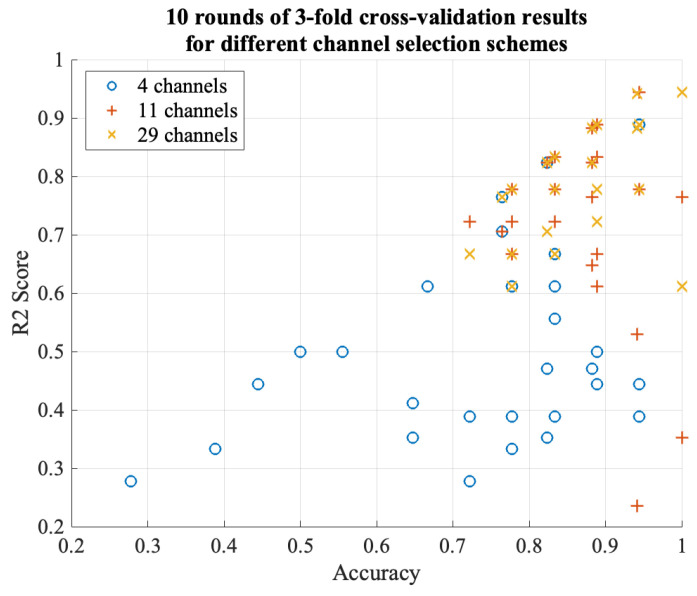
Comparison of 10 rounds of three-fold cross-validation results between different channel selection schemes on eyes-closed dataset.

**Table 1 sensors-24-04634-t001:** PD and Control Participant Demographics.

Dataset	UNM		Iowa	
**Condition**	**PD**	**Control**	**PD**	**Control**
Sex	17M 10F	17M 10F	6M 8F	6M 8F
Age	69.5 ± 8.7	69.5 ± 9.3	70.5 ± 8.7	70.5 ± 8.7
MMSE	28.7 ± 1	28.8 ± 1	-	-
MOCA	-	-	25.9 ± 2.7	27.2 ± 1.7
UPDRS	22.2 ± 10.3	-	13.4 ± 6.6	-
Year since Dx	5.7 ± 4.2	-	5.6 ± 3.2	-
EEG recording (min)	3.59 ± 1	3.63 ± 1.8	3.11 ± 1.2	3.17 ± 0.9
BDI	7.6 ± 5.3	4.8 ± 4.8	-	-
Year of Ed	17.3 ± 3.3	16.6 ± 3.1	16.6 ± 3.7	16.6 ± 2.8
Year of Ed (Parents)	12.5 ± 3.8	12.5 ± 3.1	-	-
LED (mg)	707.4 ± 448.6	-	796 ± 409	-
NAART	45.2 ± 10.3	47.1 ± 7.5	-	-

**Table 2 sensors-24-04634-t002:** Features extracted from sub-bands generated by band-pass filter.

Scales	Feature Types
Time domain	Mean, Absolute Mean, Power Mean, Standard Deviation, Kurtosis, Skewness, MAD, Interquartile Range for every Sub-Band and RMAV for Adjacent Sub-Bands.
Frequency domain	Freqeuncy Center, Mean Value of Frequency and Root Mean Square of the Frequency for Every Sub-Bands.
Entropy	Sample Entropy for every Sub-Band.

**Table 3 sensors-24-04634-t003:** Features extracted from sub-bands generated using Wavelet Packet Transform.

Scales	Feature Types
Coefficient	Mean, Absolute Mean, Power Mean, Standard Deviation, Kurtosis, Skewness, MAD, Interquartile Range of the Coefficients for every Sub-Band and RMAV for Adjacent Sub-Bands.

**Table 4 sensors-24-04634-t004:** Dimensionality of the feature vectors in different feature datasets for one participant.

Eye States	Feature Extraction Methods	Dimensionality
Eyes Closed	IIR	69,030
	Wavelet	47,790
Eyes Open	IIR	69,030
	Wavelet	47,790

**Table 5 sensors-24-04634-t005:** Overall accuracy results with all 22 features on two eyes states with different numbers of channels.

State		Models							
		**LR**	**KNN**	**SVM**	**AdaBoost**	**XGBoost**		**RF**	**ANN**
						**Tree**	**Linear**		
Eyes	29 best channels	93.75%	93.75%	**100%**	75%	87.5%	93.75%	87.5%	93.75%
closed	All 59 channels	93.75%	87.5%	93.75%	81.25%	87.5%	81.25%	87.5%	62.5%
Eyes	29 best channels	87.5%	62.5%	81.25%	75%	75%	81.25%	81.25%	**93.75%**
open	All 59 channels	81.25%	62.5%	81.25%	81.25%	75%	75%	75%	62.5%

**Table 6 sensors-24-04634-t006:** Out-of-sample Test Results.

Score Type	Eye State	LR	KNN	SVM	AdaBoost	XGBoost-Tree	XGBoost-Linear	RF	ANN	Mean
**Accuracy**	eyes open	0.857	0.714	0.821	0.821	0.821	0.821	0.786	0.821	0.808
	eyes open & closed	0.857	0.786	0.857	0.821	0.821	0.786	0.786	0.929	0.829
**Sensitivity**	eyes open	0.786	0.714	0.714	0.929	1.0	0.714	0.643	0.786	0.786
	eyes open & closed	0.714	0.623	0.714	0.929	0.929	0.642	0.571	0.929	0.754
**Specificity**	eyes open	0.929	0.714	0.929	0.714	0.643	0.929	0.929	0.857	0.830
	eyes open & closed	**1.0**	**0.929**	**1.0**	**0.714**	**0.714**	**0.929**	**1.0**	**0.929**	0.901
**AUC_ROC**	eyes open	0.841	0.783	0.832	0.796	0.827	0.821	0.849	0.821	0.821
	eyes open & closed	0.827	0.855	0.852	0.816	0.841	0.786	0.867	0.929	0.845
**R2 Score**	eyes open	0.714	0.678	0.786	0.714	0.75	0.75	0.679	0.286	0.676
	eyes open & closed	0.786	0.786	0.75	0.786	0.714	0.679	0.714	0.714	0.742

## Data Availability

The data presented in this study are openly available in Narayanan Lab, specifically in section 22: Linear predictive coding distinguishes spectral EEG features of Parkinson’s disease, at https://narayanan.lab.uiowa.edu/article/datasets (accessed on 13 May 2024).

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
