# Peer review of "Multi-Scale Feature and Multi-Channel Selection toward Parkinson’s Disease Diagnosis with EEG"

_sensors, 2024, doi:10.3390/s24144634_

Round 1
Reviewer 1 Report
Comments and Suggestions for Authors
my comments are in the attached file.

Reviewer 2 Report
Comments and Suggestions for Authors
In the paper, the authors describe new approaches to EEG analysis using machine learning methods. The topic of identifying Parkinson's disease in the early stages has not yet been resolved, and the results described in the article will make it possible to advance the study.
The authors used verified datasets of sufficient size. All necessary metrics for assessing the adequacy of the algorithm's performance, such as accuracy, sensitivity and specificity, are provided. The values ​​of these parameters indicate a fairly high level of operation of the applied algorithm. A cross-validation check was also carried out.
I would like to express my concern about the description of 100% accuracy in the abstract and discussion without clarifying that these results were obtained only when recording EEG with eyes closed.
Also I would like to kindly advise the authors to expand their conclusions a little by describing further plans and prospects for the study.
